# Combined Luteolin and Indole-3-Carbinol Synergistically Constrains ERα-Positive Breast Cancer by Dual Inhibiting Estrogen Receptor Alpha and Cyclin-Dependent Kinase 4/6 Pathway in Cultured Cells and Xenograft Mice

**DOI:** 10.3390/cancers13092116

**Published:** 2021-04-27

**Authors:** Xiaoyong Wang, Lijuan Zhang, Qi Dai, Hongzong Si, Longyun Zhang, Sakina E. Eltom, Hongwei Si

**Affiliations:** 1Department of Human Sciences, Tennessee State University, Nashville, TN 37209, USA; xiaoyong.wang@vumc.org (X.W.); lzhang3@tnstate.edu (L.Z.); geniisofla@gmail.com (L.Z.); 2Vanderbilt University Medical Center, Department of Medicine, Division of Rheumatology and Immunology, Nashville, TN 37232, USA; 3Department of Veterinary Medicine, Northwest University for Nationalities, Lanzhou, Gansu 730030, China; 4Vanderbilt Epidemiology Center, Vanderbilt-Ingram Cancer Center, Department of Medicine, Division of Epidemiology, Vanderbilt University School of Medicine, Nashville, TN 37203, USA; qi.dai@vanderbilt.edu; 5Institute of Computational Science and Engineering, Qingdao University, Qingdao, Shandong 266071, China; sihz03@126.com; 6Department of Biochemistry, Cancer Biology, Neuroscience & Pharmacology, Meharry Medical College, Nashville, TN 37208, USA; seltom@mmc.edu

**Keywords:** synergistic, breast cancer, luteolin, indole-3-carbinol, combination, estrogen receptor alpha, cyclin-dependent kinases, apopotosis, cell cycle, xenograft mice

## Abstract

**Simple Summary:**

Anti-cancer effects of bioactive compounds have been extensively investigated; however, the effective dosages of the bioactive compounds are too high to be obtained by oral intake. Our study aimed to assess if combined two bioactive compounds, luteolin (LUT) and indole-3-carbinol (I3C), at low dosages that LUT or I3C along has no significant effect, synergistically exerts anti-breast cancer. We confirmed that combined LUT and I3C synergistically suppressed estrogen receptor-alpha positive breast cancer in cultured cells and cells-derived xenograft mice. Our results also indicated two possible molecular pathways involving the synergistic effects of the combination of LUT and I3C. Our findings provide a practical approach to treat or prevent breast cancer by combining two bioactive compounds.

**Abstract:**

The high concentrations of individual phytochemicals in vitro studies cannot be physiologically achieved in humans. Our solution for this concentration gap between in vitro and human studies is to combine two or more phytochemicals. We screened 12 phytochemicals by pairwise combining two compounds at a low level to select combinations exerting the synergistic inhibitory effect of breast cancer cell proliferation. A novel combination of luteolin at 30 μM (LUT30) and indole-3-carbinol 40 μM (I3C40) identified that this combination (L30I40) synergistically constrains ERα^+^ breast cancer cell (MCF7 and T47D) proliferation only, but not triple-negative breast cancer cells. At the same time, the individual LUT30 and I3C40 do not have this anti-proliferative effect in ERα^+^ breast cancer cells. Moreover, this combination L30I40 does not have toxicity on endothelial cells compared to the current commercial drugs. Similarly, the combination of LUT and I3C (LUT10 mg + I3C10 mg/kg/day) (IP injection) synergistically suppresses tumor growth in MCF7 cells-derived xenograft mice, but the individual LUT (10 mg/kg/day) and I3C (20 mg/kg/day) do not show an inhibitory effect. This combination synergistically downregulates two major therapeutic targets ERα and cyclin dependent kinase (CDK) 4/6/retinoblastoma (Rb) pathway, both in cultured cells and xenograft tumors. These results provide a solid foundation that a combination of LUT and I3C may be a practical approach to treat ERα^+^ breast cancer cells after clinical trials.

## 1. Introduction

Endocrine therapy is the mainstay treatment of estrogen receptor alpha positive (ERα^+^) breast cancer cells, which accounts for the majority (up to 70%) of all breast cancers [1]. Endocrine therapeutic agents, including selective ER modulators, selective ER degraders, and aromatase inhibitors, provide high therapeutic efficacy by intervening in ERα-dependent growth of breast cancer cells. However, using a single targeted agent inevitably leads to breast cancer evolution during therapy to develop drug resistance, presenting a significant breast cancer treatment challenge [2]. Therefore, the development of a strategy combining ERα interfering and other different targets has been proposed for increasing tumor cell killing while reducing the risk of drug resistance.

Plant-derived phytochemicals, non-essential nutrients but with health benefits, have been reported with a wide range of anti-cancer effects through various mechanisms with very low systemic toxicity. Luteolin (3′,4′,5,7-tetrahydroxyflavone, LUT) is a flavonoid present in many commonly consumed vegetables, including thyme, Chinese celery, radicchio, and peppers [3]. Indole-3-carbinol (I3C) is a bioactive compound found in cruciferous vegetables, including brussels sprouts, cabbage, kale, cauliflower, and broccoli. Recent studies show that individual luteolin [4,5,6] and I3C [7,8,9] exerts an anti-breast cancer effect both in ER^+^ breast cancer and TNBC cells through inhibiting cell proliferation. However, the half-maximal inhibitory concentration (IC50) of individual LUT and I3C in ER^+^ and TNBC cells is 50 μM [10,11] and 200–490 μM [8,9], respectively. These high concentrations are not likely to be physiologically achievable that the highest plasma concentration of LUT, I3C, and its metabolites transiently reach 15 μM (50 μmol/kg, gastric intubation in rats) [12] and 0.04 μM (250 mg/kg, oral in mice) [13], respectively. One of our approaches to narrow the concentration gap between in vitro studies and human studies is to combine two phytochemicals to synergistically inhibit breast cancer, while the individual phytochemicals do not have an anti-tumor effect at the selected dosages.

In the present study, we screened phytochemicals to select combinations exerting a synergistic inhibitory effect on breast cancer using a cell proliferation assay. We identified that a novel combination of LUT and I3C synergistically constrains ERα^+^ breast cancer cell proliferation (MCF7 and T47D) and tumor growth in MCF7-derived xenograft mice. At the same time, the individual LUT and I3C do not have this anti-cancer effect at the selective dosages both in vitro and in vivo. This combination’s anti-proliferative effect involves inducing G1 cycle arrest and apoptosis both in MCF7 and T47D cells. Primarily, this combination of LUT and I3C synergistically downregulates two primary therapeutic targets ERα and cyclin dependent kinase (CDK) 4/6/retinoblastoma (Rb) pathway, both in cultured cells and xenograft tumors. In addition, this combination of LUT and I3C does not have toxicity in human endothelial cells than the current commercial drugs, while this combination exerts anti-tumor effects. These results suggest that the combination of LUT and I3C is an efficient approach to treat ERα^+^ breast cancer without side effects on the vasculature.

## 2. Results

### 2.1. Combination of LUT and I3C Synergistically Inhibits ER^+^ Breast Cancer Cells Growth

Twelve phytochemicals (Appendix A and Appendix A) were chosen to determine the concentration range of interest for pairwise combination screens using a cell proliferation assay. A wide range of concentrations (521-fold range) of a single agent was used to establish a dose-response curve for each of these agents using 3-day cell proliferation assays (Appendix A). As shown in Figure 1A,B, LUT and I3C dose-dependently inhibited cell proliferation in ER^+^ breast cancer cell lines MCF7 and T47D and triple-negative breast cancer (TNBC) cell lines MDA-MB-231 and BT-549. The relevant concentrations of LUT and I3C at EC_20_, EC_50_, and EC_80_ for four cell lines are shown in Figure 1C. Our primary objective is to select combinations that combined two chemicals that exert a synergistic inhibitory effect, while the individual chemical does not have a low inhibitory effect at the selected concentrations. Therefore, serial concentrations of less than 20% inhibitory effect (≤EC_20_) of the individual chemical were used for pairwise combination screens in breast cancer cells. As shown in Figure 1D, the combined LUT and I3C synergistically inhibited ER^+^ breast cancer cells MCF7 and T47D cell growth only, whereas no synergistic effects were observed in TNBC BT-549 and MDA-MB-231 cells at the ranges of the selected concentrations. The synergistic inhibitory effect of the combination was determined by the combination index (CI): CI > 1, antagonistic effect, and CI < 1, synergistic effect as described by the Chou-Talalay plot [14]. Plots of potent inhibitory effects (*Fa*) and combination index (CI) of LUT and I3C are shown in Figure 1E. The optimized combination L30I40 (LUT 30 µM + I3C 40 µM, *Fa* > 0.7, CI = 0.58 in MCF7; *Fa* > 0.6, CI = 0.56 in T47D) with a strong synergistic effect was selected for further in vitro studies.

To further evaluate the anti-cancer effect and the toxicity of our selected combination of LUT and I3C, chemotherapeutic drug doxorubicin (DOX) and ER^+^ breast cancer targeted therapeutic drug tamoxifen (TAM) were applied both in MCF7 and human endothelial cell line EA.hy926. We found that our combination (L30I40) does not affect cell viability (Figure 1F,G), but DOX killed more than 70% in EA.hy926 cells, while L30I40 and the two commercial drugs have shown similar anti-cancer effects at the relevant concentrations. This result is in line with previous studies that the DOX (0.5 µM) induced cardiotoxicity [15]. Additionally, we observed that tamoxifen (5 µM) significantly changed the morphology of EA.hy926 cells, although it did not kill these cells. Therefore, this combination (L30I40) is a practical chemotherapeutic approach to treat ER^+^ breast cancer without side effects than the two commercial drugs. These results suggest that this combination L30I40 may have a brilliant application to treat ER^+^ breast cancer in humans.

### 2.2. Combination of LUT and I3C Induces G1 Cell Cycle Arrest and Apoptosis in ER^+^ Breast Cancer Cells

Next, we explored the time course of cell growth inhibition by combining LUT and I3C in ER^+^ breast cancer cells. Surprisingly, both the MCF7 and T47D cell growth was not affected by the treatment of individual compounds or the combination within 24 h. In contrast, the combination of LUT and I3C significantly suppressed the cell growth after a 48-h treatment. Specifically, cell numbers were decreased considerably after 72 h by the combination, compared to 48 h (Figure 2A). To further understand the action of the combination on cell growth inhibition, we determined cell populations in each cell-cycle phase using propidium iodide (PI) staining. We found that the combination L30I40 significantly (*p* < 0.01) induced G1 arrest in MCF7 cells, both at 24 h and 48 h time points, while the individual LUT caused G1 arrest only at 48 h (Figure 2B). Consistent with the observation of cell number shrinkage in Figure 2A, the apoptosis analysis indicated that both LUT and L30I40 dramatically induced MCF7 cell death (Annexin V) at 48 h (Figure 2C).

To test if L30I40 induces growth inhibition and cell apoptosis through regulating cell-cycle-related proteins and apoptosis effectors in MCF7 and T47D, we measured master regulators of cell cycle p21, p53, and apoptosis effectors Bcl-Xl and Bax in cells after a 24- and 48-h treatment by immunoblotting. The results showed that the p21 protein level was increased in MCF7, T47D subjected to LUT alone and L30I40, but not I3C alone after the 48-h treatment (Figure 2D). Although p53 regulates p21 expression in the mediation of cell cycle progression, a constant level of p53 was detected in this study, indicating that the LUT and L30I40 treatment upregulate p21 through the p53-independent pathway (Figure 2D). As the observation of cell numbers decreased by L30I40 after 48 h, we next studied whether L30I40 induces apoptosis effectors. The immunoblot analysis showed that the L30I40 significantly decreased Bcl-xL protein expression in MCF7 and T47D cells when treated cells after 48 h compared to LUT or I3C alone. Moreover, Bax protein expression was increased by LUT or L30I40 at 48 h in both two ER^+^ breast cancer cells (Figure 2E). Interestingly, the combination L30I40 significantly affected p21, Bcl-Xl, and Bax protein expression in 24 and 48 h in MCF7 cells, but this combination only affected these protein levels after 48 h in T47D cells (Figure 2D,E). These results indicate that MCF7 cells are more sensitive to the combination L30I40 than T47D cells, which aligns with the individual chemicals’ EC values that T47D cells require higher concentrations (Figure 1C). Cyclin D1-CDK4/6 complexes play a central role in promoting G1-S-phase transition in ER^+^ breast cancer [16], and p21 inactivated the cyclin D1-CDK4/6 complexes to block the G1-S transition by directly binding to the complexes [17]. Intriguingly, our results showed that our combination L30I40 significantly decreased CDK4/6 protein levels, mainly LUT targeted CDK4 only, but I3C preferred CDK6 exclusively (Figure 2F). Thus, the synergistic inhibitory effect of the combination L30I40 results from the combination of the inhibition of CDK4 by LUT and the attenuation of CDK6 by I3C, which further increases the binding p21 to the cyclin D1-CDK4/6 complexes. Based on these findings, we propose that the combination L30I40 inhibits breast cancer cell proliferation by both inducing the G1 cell cycle arrest through disruption of cyclin D1-CDK4/6 complex activities and enhancing cell apoptosis through regulating protein levels of Bcl-xL and Bax (Figure 2G).

### 2.3. Combination of LUT and I3C Suppresses E_2_-Induced MCF7 Cell Growth through Modulating ERα Levels

The combination of LUT and I3C selectively targets ER^+^ breast cancer cell MCF7 and T47D, but not TNBC cell lines BT-549 and MDA-MB-231 (Figure 1D), implicating that the synergistic inhibitory effects of the combination are mediated by the ERα signaling pathway. We first investigated if individual LUT or I3C contains E_2_-stimulated MCF7 cell growth. We found that LUT (30 µM), but not I3C (40 µM), inhibited E_2_-stimulated MCF7 cell growth (Figure 3A,B). Estrogen receptor response element (ERE) luciferase results showed that the combination L30I40 contained ERα promoter activation both with and without E_2_, but a single administration of LUT worked without E_2_ (Figure 3C). Accordingly, LUT alone or combined with I3C significantly decreased the ER protein level and inhibited the phosphorylation of retinoblastoma (Rb) after the 48-h treatment (Figure 3D,E). In addition, we found that the combined LUT and I3C also synergistically reduced the expression of SIRT1 in MCF7 and T47D cells (Figure 3D), which has been proved essential for oncogenic signaling by estrogen/ERα in ER^+^ breast cancer. Consistent with the previous studies [18], we observed that the level of ERα rapidly decreased (within 120 min) upon subsequent exposure to the E_2_ treatment (Figure 3F), whereas E_2_-induced ERα decline was diminished by the depletion of estrogen in MCF7 cells. Intriguingly, the combination L30I40 decreased the ERα protein level, both in the presence and absence of estrogen (Figure 3F). Given that the primary therapeutic mechanism for a selective estrogen receptor degrader (such as fulvestrant) in ER^+^ breast cancer is the induction of ERα degradation in a proteasome-dependent manner, we next assessed whether the combination L30I40 induced rapid ERα protein decreasing via ERα degradation. Notably, pre-treated MCF7 cells with 10 µM proteasome inhibitor MG132 remarkedly (*p* < 0.01) reversed the reduction of ERα and SIRT1 by the combination treatment for 24 h (Figure 3G), suggesting that the combination L30I40 downregulates ERα and SIRT1 levels depending on the activity of the proteasome.

### 2.4. Combination of LUT and I3C Suppresses MCF7 Xenograft Tumor Growth

To confirm if the combined LUT and I3C also synergistically inhibit tumor growth in animal models, xenograft breast cancer mice were generated by injecting MCF7 cells in the mammary fat pad immunodeficient NOD scid gamma (NSG) mice. Treatments (intraperitoneally, i.p.) were given at week 3 after tumor inoculation for 5 weeks (Figure 4A). We found that the combination treatment of LUT (10 mg/kg/day) and I3C (20 mg/kg/day), but neither LUT nor I3C alone, significantly (*p* < 0.05) decreased the tumor size and volume compared to the control treatment (Figure 4B,D) without effects on body weight (Figure 4C). Moreover, there were no significant changes in the weight, color, and other anatomical changes of the brain and reproductive tract (data not shown). As expected, LUT or the combination treatment markedly reduced the tumor weight, and the combination group has a significantly lower tumor weight than the tumor weight in LUT alone mice (Figure 4E). Moreover, the combination of LUT and I3C significantly reduced the ERα, CDK4/6 protein levels of tumors (Figure 4F), which is in line with our in vitro results of ERα and CDK4/6 (Figure 2 and Figure 3). These results confirm that the combination of LUT and I3C synergistically inhibits ERα positive breast cancer through regulating ERα and CDK4/6/Rb pathways both in vitro and in vivo.

## 3. Discussion

Our selected combination of L30I40 synergistically inhibited cell proliferation only in ER^+^ breast cancer MCF7 and T47D cells, not in TNBC BT-549 and MDA-MB-231 cells, suggesting that this synergistic inhibitory effect requires the ERα pathway. Indeed, our results show that the combined LUT and I3C suppressed ERα protein expression profoundly both in cultured MCF7 and T47D cells and MCF7-derived xenograft tumors. This is further supported by the fact that the combination L30I40 synergistically inhibited ERα gene expression promoter activity and degraded ERα protein rapidly. These results align with previous studies that the individual LUT decreases ERα protein by inhibiting ERα gene expression in ERα^+^ breast cancer cells [19], I3C triggers aryl-hydrocarbon receptor (AhR)-dependent ERα protein degradation [20], and the combined LUT and I3C synergistically reduced ERα mRNA level in MCF7 cells [21]. Moreover, we found that this combination L30I40 attenuated the protein expression of sirtuin 1 (SIRT1), an NAD+-dependent class III histone deacetylase linked to gene silencing, control of the cell cycle, and apoptosis, which is required for ERα-mediated gene transcription and cell proliferation [22,23]. Therefore, the combination of LUT and I3C exclusively suppresses ER^+^ breast cancer synergistically via regulating the SIRT1/ERα pathway. Interestingly, our other combination of LUT and curcumin (CUR) synergistically contains TNBC only, but not ER^+^ breast cancer (separate article). To understand the mechanism of these two combinations with LUT in common, why LUT+I3C exclusively inhibits ER+ breast cancer, but LUT+CUR only synergistically suppresses TNBC, we further confirmed that this combination of LUT and I3C that reduced ERα protein can be abolished by the specific proteasome inhibitor MG132 in MCF7 cells. However, the CUR-degraded ERα protein in ERα^+^ breast cancer cells cannot be reversed by MG132 [24]. These results indicate that I3C maybe the major inhibitor of ERα signaling by the combination of LUT and I3C, and CUR may be the major contributor of TNBC inhibition by the combination of LUT and CUR, given that the anti-ER^+^ breast cancer effect of CUR was not mediated by the AhR-degraded ERα pathway [24].

The elevated endogenous estrogen (estradiol or E_2_) level is one of the critical risk factors of breast cancer development [25], which is supported by the fact that bilateral oophorectomy significantly reduced odds of breast cancer (odds ratio = 0.74) [26]. The normal range of plasma E_2_ level is 30–400 pg/mL in pre-menopausal women, whereas it falls below 30 pg/mL in post-menopausal women. Interestingly, many studies [27,28] reported that the median plasma level of E_2_ is around 10 pg/mL (approximately equal to 30 pM) in post-menopausal ERα^+^ breast cancer patients, and intra-tumoral E_2_ levels are 10 to 20 times higher than those in plasma. Therefore, it is critical to know the impacts of E_2_ levels on cell proliferation’s inhibitory effects by combined or individual LUT and I3C. While the combination L30I40 significantly inhibited MCF7 cell growth both with and without E_2_, LUT has a multiple-phasic impact on MCF7 cell proliferation. A high concentration of LUT (30 µM) greatly exerted an anti-proliferative influence on MCF7 cells in the presence of both low levels (less than 10 pM) and excess E_2_ (1–10 nM). However, intermediate concentrations of LUT (8–16 µM) slightly promote MCF7 cells proliferation only when E_2_ is absent or at a concentration lower than 1 pM, whereas LUT at 8–16 µM did not promote MCF7 cells growth at concentrations (10 pM–1 nM) of E_2_, the level of plasma, and intra-tumoral E_2_ in post-menopausal women with ERα^+^ breast cancer (Appendix A). On the contrary, I3C showed mono-phasic effects on the inhibition of MCF7 cell growth when exposed to ranges of physiological-related E_2_ levels, suggesting that I3C works through different mechanisms on the suppression of proliferation in MCF7 cells. These data may help us understand the mechanism with LUT in common, why LUT+I3C exclusively inhibits ER+ breast cancer, but LUT+CUR only synergistically suppresses TNBC, both in cells and xenograft mice.

Promoting cell cycle arrest and apoptosis are effective approaches in inhibiting cancer cell proliferation. Due to the critical roles of the CDK4/6/Rb pathway in cell cycle arrest and apoptosis, CDK4/6 inhibitors have emerged as promising candidates for cancer treatment. In the present study, the combination L30I40 significantly arrested MCF7 cells in the G1 cell phase, which is accompanied by downregulated CDK4/6 protein levels, Rb phosphorylations at Ser780 and Ser795, as well as increased p21 expression, the primary molecules of the CDK4/6/Rb pathway. Indeed, upon activation by upstream signals, the CDK4/6 is active to phosphorylate Rb, thereby promoting dissociation of the transcriptionally repressive Rb-function of the E_2_ family (E_2_F) complex. The released E_2_F transcription factors are then free to activate genes required for entry into the S phase and DNA replication [29]. In addition, a recent study reported that p21 levels were proportional to CDK4/6 inhibitor sensitivity and had the potential as a monitoring marker for ribociclib, an FDA-approved CDK4/6 inhibitor in late-stage breast cancer [30]. Notably, while our combination L30I40 significantly decreased CDK4/6 protein levels, we found that LUT targeted CDK4 only, but I3C preferred CDK6 exclusively, indicating that the synergistic inhibition of CD4/6 complex results from the combination of the inhibition of CDK4 by LUT and the suppression of CDK6 by I3C. Moreover, CDK6 has directly involved in transcription in tumor cells and hematopoietic stem cells in addition to the kinase activity with CDK4 in cell-cycle progression [31]. Additionally, our combination also induced MCF7 cell apoptosis and decreased Bcl-xL, and increased Bax protein expression. These results are in line with previous studies that the individual LUT [32] and I3C [33] exert this cell cycle arrest and pro-apoptotic outcomes at 50 and 100 µM, respectively, which are much higher than the concentrations of LUT (30 µM) and I3C (40 µM) used in this study. Significantly, the individual LUT30 or I3C40 did not have such pro-apoptotic and pro-cycle arrest effects, while the combination L30I40 has these effects. These results indicate that the synergistic effects of combination L30I40 on cell cycle arrest and apoptosis are not a simple accumulation of LUT and I3C, resulting from sophisticated mechanisms.

Based on the increasing studies, we recently summarized five mechanisms to understand how a combination of two or more phytochemicals exerts synergistic effects in cells, animals, and humans [34]: (1) Enhance the bioavailability of phytochemicals; (2) increase antioxidant capacity; (3) interact with gut microbiome (change microbial profiles, reduce endotoxin, and increase gut integrity); (4) target the same and/or different signaling pathway; and (5) apply two or more of these four mechanisms simultaneously. For instance, AhR exerts the proteasome-dependent degradation of ERα proteins by binding its ligands, including 2-, 3-, 7-, 8-tetrachlorodibenzo-p-dioxin in MCF-7 human breast cancer cells [35]. While LUT is well-known as an AhR antagonist, I3C can also bind to AhR to translocate its receptor from the cytosol to the nucleus [36] to further decrease the ERα mRNA level in MCF7 cells [21]. Therefore, the accumulation of the anti-AhR activity from LUT and I3C contributes to the ERα level and then inhibits ER^+^ breast cancer cell growth. While the I3C-reduced ERα expression can be rescued by a specific proteasome inhibitor MG132 [20], to our knowledge, there is no data that MG132 can reverse LUT-induced ERα expression, suggesting that I3C may be the major contributor to the combination of LUT and I3C attenuated ERα protein expression in cells and tumors. Given that the combination L30I40 reduced ER-a protein expression was reversed by MG132. On the contrary, CUR-reduced ERα expression cannot be rescued in T47D ERα^+^ cells [24]. These differences in the AhR/ERα pathway between I3C and CUR may contribute to the mechanism that with LUT in common, why LUT+I3C exclusively inhibits ER^+^ breast cancer, but LUT+CUR only synergistically suppresses TNBC, both in cells and xenograft mice. Moreover, LUT can inhibit the AhR activity both in MCF7 cells and mouse Hepa-1 cells, but other food-derived AhR antagonists such as genistein, daidzein, and diosmin only work in Hepa-1 cells [37]. Another example is that the synergistic inhibition of CD4/6 complex results from the combination of the inhibition of CDK4 by LUT and the suppression of CDK6 by I3C. In addition, the I3C-affected CDK6 can regulate cancer cell proliferation both by arresting the cell cycle and involving the transcription of relevant genes. Therefore, the interactions between phytochemicals on multiple levels (kinase activity and transcription) and pathways contribute to the synergistic effects when two or more phytochemicals are combined, which deserves more investigations to understand why combined phytochemicals work better than individual ones and to support the concept that food diversity is good for human health.

Although the combined LUT and I3C exerted synergistic anti-cancer effects both in vitro and in vivo studies, the concentrations of LUT (30 µM) and I3C (40 µM) used in our in vitro studies are still much higher than the plasma concentrations of LUT and I3C in humans, a limitation of the present study. However, our selected concentrations of LUT and I3C are much lower than the concentrations of LUT (50 µM) [32] and I3C (100 µM) [33] in similar studies. Moreover, these two concentrations are the optimum levels based on our criteria that the individual chemical has no or very low effect; in contrast, the combination of LUT and I3C has a synergistic inhibitory effect (Fa > 0.7, growth inhibition > 70%) with the lowest combination index (CI = 0.58). Actually, this is the first step to lower the concentration of the chemicals while having anti-cancer effects, and we will further reduce the dosages of the substances by combining more chemicals or other approaches shortly. Another limitation of this study is that external 17 β-estradiol was supplemented to mice throughout the experiment due to the critical roles in engraftment and prevention of urinary tract complications [38]. To minimize the effects of external 17 β-estradiol, we supplemented 17 β-estradiol to mice only in the first week in the nest studies since a recent study reported that 1 week of external 17 β-estradiol supplementations could support the tumor establishment and prevent urinary complications [39].

## 4. Materials and Methods

### 4.1. Cell Culture and Treatment Reagents

All the human cell lines used in this study were obtained from the American Type Culture Collection (ATCC). The ERα^+^ breast cancer cell line MCF7, T47D, and human endothelial cell line EA.hy.926 were maintained in DMEM. Triple-negative breast cancer cell lines MDA-MB-231 and BT-549 were maintained in RPMI 1640. The medium was supplemented with 10% fetal bovine serum (FBS) and 1% penicillin/streptomycin. To evaluate the effect of selected dietary phytochemicals on E_2_-induced ERα degradation, MCF7 was switched to phenol red-free DMEM with 10% charcoal-stripped FBS and 1% penicillin/streptomycin for a 48-h hormone starvation, then treated by different phytochemicals (Sigma, St. Louis, MO, USA, Appendix A) with or without 17 β-estradiol (Cayman). Luteolin and I3C were from Sigma-Aldrich. Doxorubicin, tamoxifen, fulvestrant, and MG132 were purchased from Selleckchem. All the chemicals were dissolved in dimethyl sulfoxide (DMSO) and used at multiple indicated concentrations.

### 4.2. Cell Viability and Dose-Response Curve Fitting

Breast cancer cells (MCF7, T47D, MDA-MB-231, and BT-549) were seeded in 12-well, 24-well or 96-well plates at 10%~20% confluency in a medium with 10% FBS and 1% penicillin/streptomycin. After overnight incubation, cells were treated with different concentrations of phytochemicals (1–512 µM, 2-fold dilution series) and returned to the CO_2_ incubator for the measurement of cell viability at different time points (24, 48, and 72 h) using a metabolic assay (CellTitre-Glo, Promega or WST-1, Roche, Basel, Switzerland). Data are analyzed to generate a dose-response curve (nonlinear curve fitting) and growth rate inhibition (GRI). The concentrations at 20% (EC_20_) or 50% (EC_50_) response were determined by OriginPro.

### 4.3. Pairwise Combinations Screening

A mixture of the highest concentration (≤EC_20_) of the individual compounds at a ratio of 1:1, followed by a non-constant fold reduced concentration, was used for screen optimum combinations. The DMSO concentration was normalized for each well. After a 72-h treatment, cell viability was measured as described above. Experiments were performed at least three times, and each combination pair had two duplicate wells for each independent experiment. The growth rate inhibition (GRI) was normalized to a vehicle control. The mean value of GRI for each combination pair was displayed as a combination matrix plot. The synergistic effect of combination pairs was evaluated by the *Fa*-CI plot (CI plot or Chou-Talalay plot [14]). The combination index (CI = D_1_/D_x1_ + D_2_/D_x2_) > 1 indicates an antagonistic effect, CI = 1 indicates an additive effect, and CI < 1 indicates a synergistic effect. The optimized doses of combination pair with high effect levels (*Fa* > 0.7) and low CI (<0.6) values were used for further studies.

### 4.4. Cell Cycle Assay

Cells were seeded in 60 mm tissue culture dishes and incubated overnight. After treatment with an optimized dose of the individual compound or the combination for a 24 or 48 h period, cells were collected then fixed in 70% ice-cold ethanol at 4 °C overnight. Cells were washed with ice-cold PBS and then stained with propidium iodide (PI) supplemented with DNase-free RNase A (FxCycle^TM^ PI/RNase, Invitrogen, Carlsbad, CA, USA) for 30 min at room temperature and analyzed by flow cytometry.

### 4.5. Cell Apoptosis Assay

Cell apoptosis was quantified with the Annexin V Apoptosis Detection Kit (eBioscience, San Diego, CA, USA). After treatment with the optimized concentration of individual compounds or combination pairs for 24 or 48 h, cells were harvested and suspended in a binding buffer. Fluorochrome-labeled annexin and propidium iodide was used to stain cells for 30 min at room temperature, then analyzed by flow cytometry.

### 4.6. ERE Luciferase Reporter Assay

MCF7 cells were seeded in a 24-well plate at 20% cell confluency and were incubated in an E_2_ depletion phenol-red free DMEM medium (2% charcoal-stripped FBS) for 48 h. Cells were treated with an optimized dose of the individual compound in 10% charcoal-stripped FBS contained phenol-red free DMEM medium with or without 1 nM E_2_ for 48 h, and cells were subsequently transfected with a 3ERE-luciferase plasmid (Addgene) using the Lipofectamine 3000 transfection reagent (Thermo Fisher Scientific, Waltham, MA, USA). After 24 h transfection, cells were lysed in a reporter lysis buffer (Promega, Madison, WI, USA), and total protein concentration was determined by the BCA protein assay kit (Pierce, Pittsburgh, PA, USA). Equal total proteins of each sample were used for the luciferase assay with the Luciferase Assay System (Promega). After 10 min incubation at 37 °C, luminescence was recorded and analyzed using the BioTek Cytation 3 plate reader.

### 4.7. Immunoblotting, Immunoprecipitation, and Immunofluorescence Staining Experiments

Cells or homogenized tissues were lysed in a RIPA buffer (25 mM Tris-HCl pH 7.6, 150 mM NaCl, 1% NP-40, 1% sodium deoxycholate, 0.1% SDS) supplemented with protease inhibitor MG132 (Selleckchem, Houston, TX, USA) and phosphatase inhibitors (Roche, Basel, Switzerland). Total protein concentrations were determined by the BCA protein assay kit (Pierce). For immunoprecipitation experiments, 1 mg (200 µl) of protein lysates was used for immunoprecipitation with 20 µL anti-cyclin D1 (2922, cell signaling) antibody conjugated to Protein A Magnetic Beads (10016D, Invitrogen). The same amount of normal rabbit IgG-conjugated magnetic beads (DA1E, cell signaling) was used for the control. Beads were collected by a magnetic rack, and then the proteins were denatured at 95–100 °C for 5 min, then followed by an immunoblotting assay with the antibodies listed below. For immunoblotting experiments, an equal amount (20–40 µg) of cells or tissue lysates were separated on SDS-PAGE and transferred to nitrocellulose membranes (GE Healthcare, Life Sciences, Chicago, IL, USA). Antibodies against the following proteins were used for immunoblotting: Cyclin D1 (2922), p21 (12D1), CDK4 (D9G3E), p53 (7F5/9282), Bcl-xL (2764), Bax (D2E11), ERα (D8H8), SirT1 (D739), GAPDH (D16H11), β-actin (13E5), HRP-linked anti-rabbit IgG antibodies (7074), IRDye^®^800CW conjugated anti-rabbit IgG (926-32211, LI-COR Biosciences), and IRDye^®^800CW conjugated anti-mouse IgG (926-32210, LI-COR Biosciences, Lincoln, NE, USA). All antibodies were purchased from Cell Signaling Technology unless indicated otherwise. The immunoreactivity was detected using either the Odyssey^®^ CLx imaging system (LI-COR) or X-film, followed by band intensities analysis in ImageJ.

For immunofluorescence staining assays, cultured MCF7 cells were fixed by 4% paraformaldehyde and then stained by appropriate fluorescent reagents, as previously described [40]. Images were captured using a Nikon TE300 fluorescence microscope.

### 4.8. Xenograft Mice Study

Five-six weeks old female NOD.Cg-*Prkdc^scid^ Il2rg^tm1Wjl^*/SzJ mice were purchased from Jackson Lab (005557 NSG) and supplemented with 17 β-estradiol in drinking water (8 µg/mL) throughout the whole experiment (water was changed three times per week). ER^+^ breast cancer cells MCF7 were maintained in T-75 flash to reach 80% confluence. Harvested cells in DMEM were mixed with a Growth Factors Reduced Matrigel Matrix High Concentration (354263, Corning) at 1:1 ratio to obtain the cell mixture containing 1 × 10^7^ MCF7 cells per mL. In addition, 100 µl (1 × 10^6^) cells mixed with Matrigel were injected at the left inguinal mammary fat pad to induce tumors. Tumor growth was measured in two dimensions (length diameter, LD, and width diameter, WD) twice each week. The volume in mm^3^ was calculated by the formula: Tumor volume (Tv) = LD × WD^2^ × 3.14/6 [41]. After 3 weeks of inoculation, all the mice with a tumor volume that reached 50–100 mm^3^ were randomly allocated into four groups (8 mice/group). Vehicle (5% DMSO, 5% Tween, 90% PBS, *v*/*v*%), luteolin (10 mg/kg body weight/day), indole-3-carbinol (20 mg/kg body weight day), the mixture of luteolin (10 mg/kg body weight/day), and indole-3-carbinol (20 mg/kg body weight/day) were administrated by intraperitoneal (IP) injection every other day for 5 weeks. The tumor size was continuously measured as described above twice each week. All the compounds were dissolved in the vehicle (5% DMSO, 5% Tween, 90% PBS, *v*/*v*%) and stored at −20 °C. At the end of the experiment, all the mice were euthanized by CO_2_ to collect the tumors and other tissues. The tumor weight and size were measured immediately. Tumors and other tissues were analyzed, stored in 10% neutral buffered formalin or −85 for future analyses. Animal experiments were approved by the Institutional Animal Care and Use Committee at Tennessee State University (Nashville, TN, USA), protocol No. 16-11-636, dated 3 March 2017.

### 4.9. Statistical Analysis

Data presentation and statistics were generated using the OriginPro software. The data shown in this study are represented as the mean ± SEM, and significance was determined by the analysis of variance (One-Way ANOVA) with Bonferroni Post-hoc tests followed by the unpaired two-tailed Students’ *t*-test for experiments with more than two different conditions. *p*-value < 0.05 was considered as statistically significant (* *p* < 0.05, ** *p* < 0.01, *** *p* < 0.001).

## 5. Conclusions

In summary, we reported for the first time to our knowledge that a combination of LUT and I3C, at a low level without effect alone, synergistically suppresses cell proliferation only in ERα^+^ breast cancer (MCF7 and T47D), but not in TNBC cells. This combination L30I40 demonstrated no toxicity in human endothelial cells compared to the current commercial breast cancer drug. Moreover, this L30I40 synergistic anti-breast cancer effect by the combination was mediated by reducing the CDK4/6/Rb pathway to induce stringent growth arrest in the G1 cell cycle, as well as induced apoptosis. This combination of LUT and I3C also synergistically contains tumor growth in MCF7 cells-derived xenograft mice. We found that the combination of LUT and I3C degrades ERα and Sirt1 proteins both in MCF7 cells and tumors. These results indicate that combining two or more phytochemicals at a relatively low level can exert synergistic anti-breast cancer effects. These results provide a solid foundation that a combination of LUT and I3C may be a practical approach to treat ERα^+^ breast cancer patients after clinical trials.

## Figures and Tables

**Figure 1 cancers-13-02116-f001:**
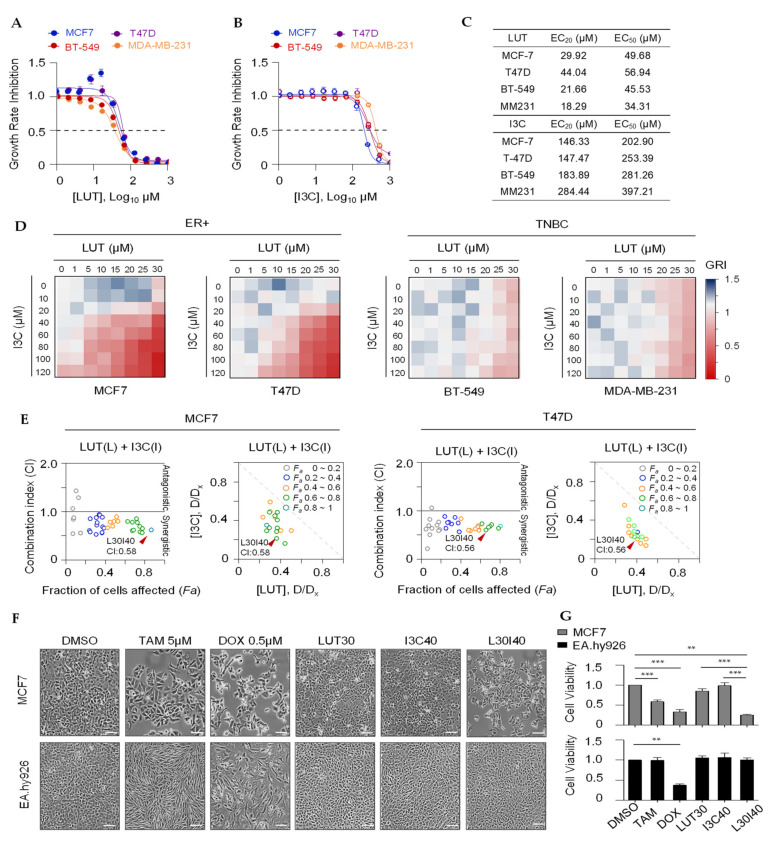
Combined luteolin (LUT) and indole-3-carbinol (I3C) synergistically inhibits estrogen receptor-positive (ER) breast cancer cell growth. (**A**,**B**) Inhibitory effects(72 h) of LUT or I3C on breast cancer cells ER+breast cancer cells MCF7and T47D and TNBCs BT-549 and MDA-MB-231 viability. (**C**) Means of ECand ECof LUT and I3C in four different breast cancer cells. (**D**) Heat map depicts potent inhibitory effects of pairwise combinations usinglow concentration (≤EC) in four breast cancer cells. (**E**) Quantitative plots Fa-CI optimize doses of LUT and I3C in combination regimes in MCF7 and T47D cells. Each dot represents on combination treatment group. Fa(Fraction of cells affected, growth inhibition rate); Combination Index(CI)< 1 indicates a synergistic effect. (**F**) Images of MCF7 and EA.hy926 cells treated with TAM, DOX, LUT, I3C, and L30I40 (concentration unit: μM)for 72 h. The scale bar in the figure above represents 50 μm. (**G**) Bar graph depicts growth inhibition of individual LUT, I3C, or combination L30I40 as well as TAMand DOX in MCF7 cells and EA.hy926 cells for 72 h. Data are means ± SEM of at least three independent experiments performed in duplicate. * *p* < 0.05; ** *p* < 0.01; *** *p* < 0.001.

**Figure 2 cancers-13-02116-f002:**
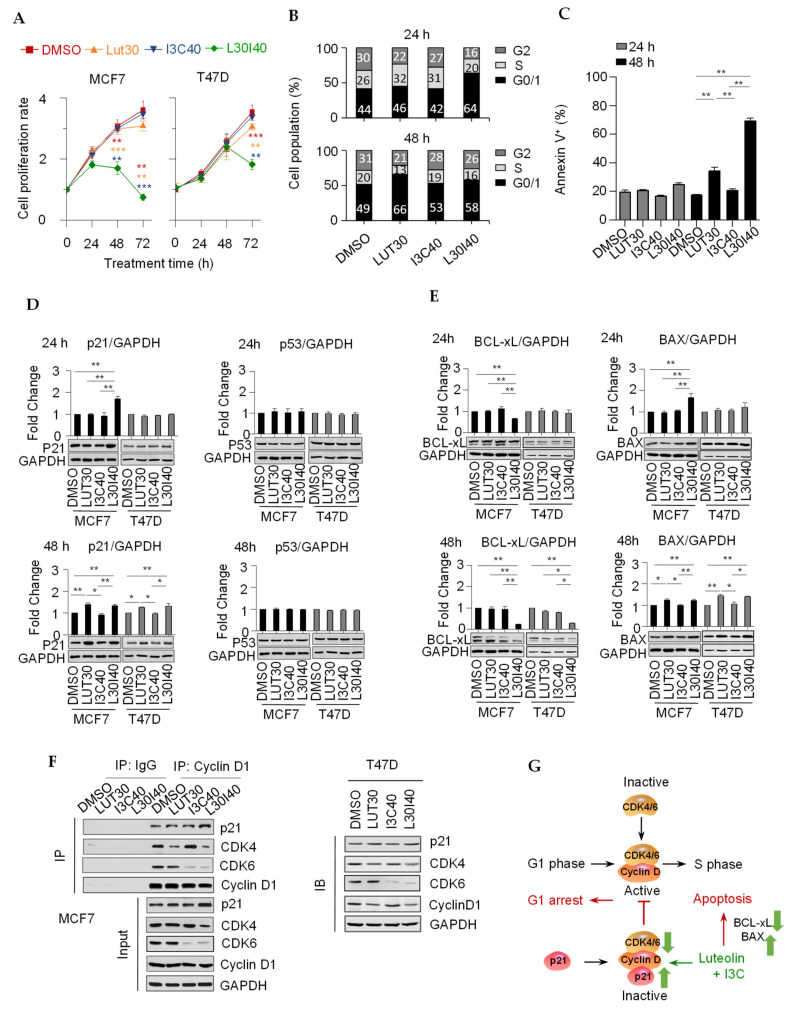
A combination of LUT and I3C induces G1 cycle arrest and apoptosis in ER+ breast cancer cells. (**A**) The time course (24 h, 48 h, and 72 h) of cell growth inhibition by individual or combination of LUT and I3C in MCF7and T47D cells. (**B**) MCF7 cells were arrested in different stages of cell cycle (percentage indicated) by the individual or the combination of LUT and I3C at 24 h and 48 h. Cycle phases were analyzed by flow cytometry after propidium iodide (PI) staining. (**C**) Combined LUT and I3C significantly induced MCF7 cell apoptosis at 48 h. Apoptotic cells were counted by flow cytometry after Annexin V staining. (**D**) Cell cycle main regulator p21 protein level (measured by Immunoblot) was significantly increased by L30I40 in MCF7cellsat 24 h and 48hbut only at 48 h inT47D cells. The MCF7 and T47D cells at 24h and 48h were the same samples on the same membrane and share the same GAPDH bands, after detecting one protein, the membrane was stripped and reprobed the second or third proteins. (**E**) Apoptotic effector Bcl-xL and Bax protein levels (by Immunoblot) were significantly changed by L30I40 at 24 h and 48 h treatment in MCF7 and T47D cells. (**F**) L30I40 decreased CDK4/6 in complex with Cyclin D1 (by immunoprecipitation assay) in MCF7 and T47D cells. (**G**) Proposed mechanism of the G1 cell-phase arrest induced by the combination of LUT and I3C. Data are means ± SEM of at least three independent experiments performed in duplicate. * *p* < 0.05; ** *p* < 0.01.

**Figure 3 cancers-13-02116-f003:**
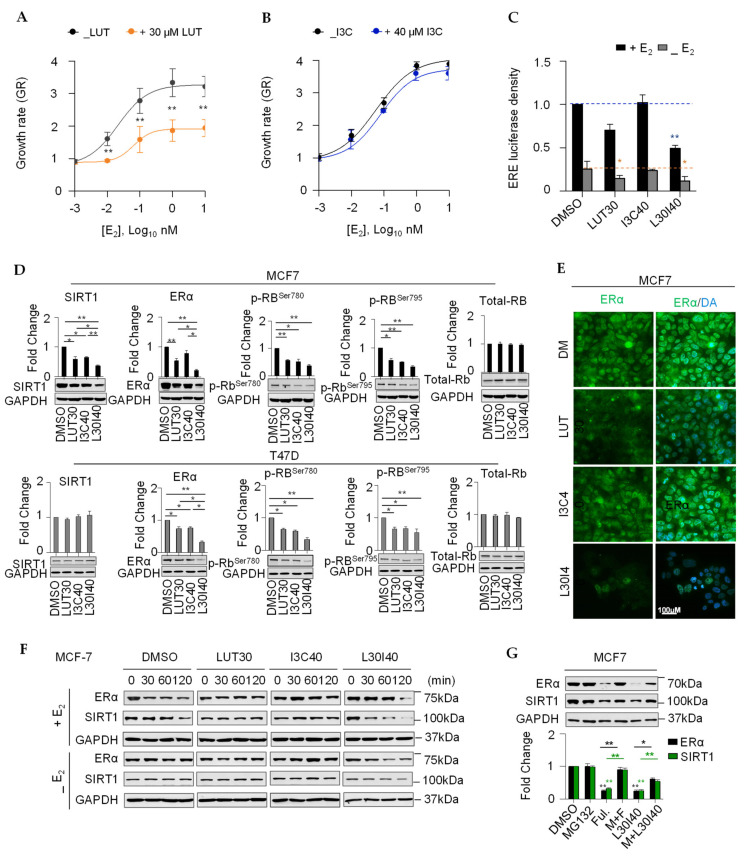
A combination of LUT and I3C suppresses E2-induced breast cancercell growth and promotes ERα degradation. (**A**,**B**) LUT (30 μM) not I3C (40 μM) inhibited E2-induced MCF7 cell growth at 72 h. (**C**) Effect of the combination L30I40 on MCF7 cells ERE activity in both E2 containing (+E2) and E2 depletion (-E2) medium. (**D**) Erα,SIRT1, and Rbprotein levels were synergistically reduced by combination L30I40 after 48h in MCF7and T47Dcells. (**E**) Immunofluorescence of ERα in MCF7 in response to LUT30, I3C40, and the combination of L30I40, respectively. (**F**) Short time (120 min) treatment of combination L30I40 synergistically degrades ERα and SIRT1 protein in MCF7 cells with or without E2(1nM). (**G**) Proteasome inhibitor MG132 partially restored L30I40-reduced ERα and SIRT1 protein levels in MCF7 cells. MG132 10 μM was used to pre-treat MCF7 cells for 30 min ahead treatment using selective estrogen receptor degrader agent. * *p* < 0.05; ** *p* < 0.01.

**Figure 4 cancers-13-02116-f004:**
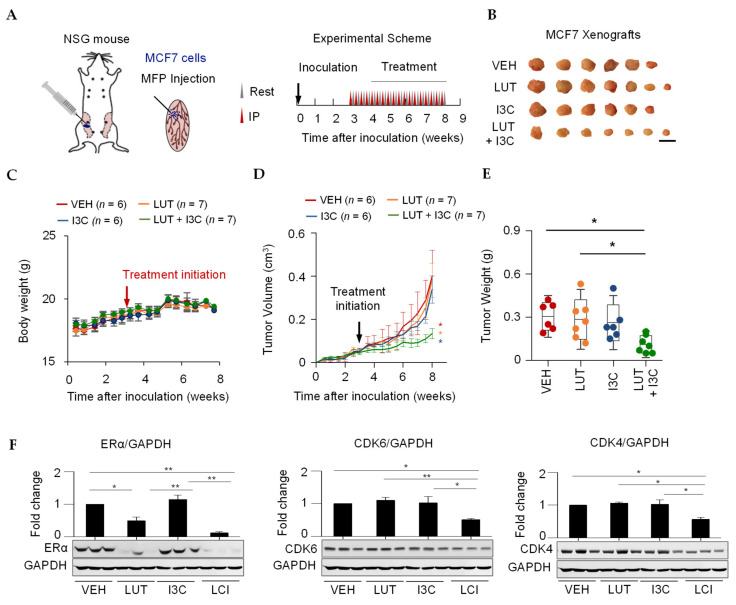
Combined LUT and I3C suppress breast cancer tumor growth in MCF7 cells-derived xenograft mice. (**A**) Schematic experimental design of MCF7 xenograft mice generation and treatments. MCF7 cells were mixed with Matrix-gel and implanted into the mammary fat pad. When tumor size reached about 50-100 mm3, xenograft mice were randomly grouped to receive vehicle (5% DMSO, 5% Tween 20, 90% PBS), LUT (10 mg/kg/day), I3C (20 mg/kg/day), or the combination of LUT and I3C (LUT 10mg/kg/day + I3C 20mg/kg/day) by i.p injection every other day for 5 weeks. (**B**) Tumors from all groups. Scale bar represents 1 cm. (**C**) No effects on body weight. Combined LUT and I3C synergistically inhibited tumor volume/size (**D**) and weight (**E**) in MCF7 xenograft mice. (**F**) Combined LUT and I3C synergistically inhibited tumor Erα, CDK6, and CDK4 protein expression. Target protein expression was normalized by GAPDH. The VEH, LUH, I3C and LCI were the same samples on the same membrane and share the same GAPDH bands, after detecting one protein, the membrane was stripped and reprobed the second or third proteins. Data are means ± SEM of the animals. N = 6–7. * *p* < 0.05; ** *p* < 0.01.

## Data Availability

The data presented in this study are available in this article and Appendix A.

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
