# Peer review of "Combined Luteolin and Indole-3-Carbinol Synergistically Constrains ERα-Positive Breast Cancer by Dual Inhibiting Estrogen Receptor Alpha and Cyclin-Dependent Kinase 4/6 Pathway in Cultured Cells and Xenograft Mice"

_cancers, 2021, doi:10.3390/cancers13092116_

Round 1

Reviewer 1 Report

The objective of this study is to screen different phytochemicals to identify a combination which could exert synergistic inhibitory effect on breast cancer cells and xenografts. The authors have reported that combination of LUT+I3C have exerted synergistic inhibitory effect on ER+ breast cancer cell line MCF-7 cells and xenografts but not on TNBC cell lines. The authors have addressed some concerns, but the manuscript could not be considered as the authors responses still do not address some of the concerns. The findings are interesting, but the manuscript require significant changes and could not be considered for publication in the current form for following reasons.

1) The manuscript is poorly written. The authors responses to previous comments were not convincing and need to make significant changes to the manuscript. In addition, as stated in earlier comments, the results are not clearly presented, and conclusions are not supported by the results. Following are few examples of discrepancies in results and conclusions.    

2) The authors have reported that p21 is significantly upregulated in MCF-7 and T-47D cells treated with L30I40 at 24h and 48h in Figure 2D legend. However, the data did not show upregulation of p21 in T-47D cells treated with L30I40 at 24h. Could authors please review the data and correct their conclusions accordingly.

3) The authors have reported that L30I40 inhibits breast cancer cell proliferation by inducing G0/G1 cell cycle arrest. However the percentage of cells arrested at G0/G1 phase in MCF-7 cells treated with L30I40 were significantly downregulated compared to cells treated with LUT30 alone at 48h. If authors conclusions were correct, the percentage of cells arrested at G1 phase in L30I40 group should at least be higher than other treatment groups. Could authors please look into it and comment on it. 

4) The authors have concluded that L30I40 arrests cells at G0/G1 phase through disruption of Cyclin D-1/CDK4/6 complex. However these effects were not noticed in MCF-7 cells treated with LUT30 alone as per the data shown in Figure 2F and yet MCF-7 cells treated with LUT30 alone have highest percentage of cells arrested at G0/G1 phase compared to L30I40 group (Fig 2B). Could authors please re-evaluate their conclusions and discuss accordingly.

5) The effect of LUT30 on cell cycle regulation noticed at 48h is not observed at 24h. If we were to correlate the expression of p21 as the reason for noticed effect of LUT30 on cell cycle regulation, it could not be attributed to LUT30 treatment as LUT30 did not regulate p21 levels at 24h.  In addition the doubling rate of MCF-7 cells is typically 24h. If LUT30 could regulate p21 levels and cell cycle dynamics like the data shown in Figure 2, the authors must have noticed the effects at 24h as well. Could authors please re-examine their conclusions and comment on it.

6) The authors have reported that the mechanism of action for LIC on tumor regression is through regulation of ER-alpha, CDK4/6 and is in line with their in-vitro findings. The expression levels of CDK-6 protein is significantly downregulated in MCF-7 cells treated with I3C and L30I40 groups. However the in-vivo data (Fig 4F) did not show correlation with in-vitro data (Fig 2F) on CDK-6 expression levels in I3C alone group vs I40 treated cells. The conclusions are not consistent with the data. Could authors please be careful with their conclusions.

Reviewer 2 Report

The authors have made substantial improvements to the manuscript. The in vivo data is backed up by solid in vitro data that provides mechanistic insights. The authors have addressed my raised issues satisfactory. I have no further critique.

Round 2

Reviewer 1 Report

The objective of this study is to screen different phytochemicals to identify a combination which could exert synergistic inhibitory effect on breast cancer cells and xenografts. The authors have reported that combination of LUT+I3C have exerted synergistic inhibitory effect on ER+ breast cancer cell line MCF-7 cells and xenografts but not on TNBC cell lines. The authors have addressed most of the concerns and the manuscript could be considered for publication.

This manuscript is a resubmission of an earlier submission. The following is a list of the peer review reports and author responses from that submission.

Round 1

Reviewer 1 Report

The objective of this study is to screen different phytochemicals to identify a combination which could exert synergistic inhibitory effect on breast cancer cells and xenografts. The authors have reported that combination of LUT+I3C have exerted synergistic inhibitory effect on ER+ breast cancer cell line MCF-7 cells and xenografts but not on TNBC cell lines. It is interesting data but the manuscript could not be considered for publication for following reasons.

Major points:

1) Could authors please explain the reason for selection of only one ER+ cell line? If the authors have performed these studies on multiple ER+ cell lines, please include the data from their findings.
2) The data from all the studies included in the manuscript have shown I3C alone did not have any effect on cell proliferation, cell cycle dynamics, cell viability, apoptosis, E2 stimulated cell growth in MCF-7 cells however the combination treatment of LUT+I3C has dramatic effect on aforementioned dynamics on MCF-7 cells. Could authors please explain how I3C which did not elicit any effect as single agent (at tested concentrations) was able to have dramatic effects in combination treatment of LUT+I3C (at the same concentrations?
3) Did authors examine the effect of LUT30 or I3C40 treatment either alone or in combination on levels of caspases to further delineate the mechanism of apoptosis and please include the data as well.
4) Could authors please run statistical analysis on data for all experiments and include the details in respective figures.
5) Could authors please indicate percentage of cells in each phase of the cell cycle in MCF-7 cells treated with LUT30 or I3C40 either individually or in combination.
6) The authors have concluded that L30I40 combinational treatment arrested MCF-7 cells in G1 phase at 24h and 48h. The data indicate slight increase in percentage of cells arrested at 48h compared to other treatment groups. Could authors please confirm whether the findings are statistically significant?

Minor points:

1) Could authors please include the number of cells seeded instead of generalized statement of cells seeded at 20% confluency.
2) Please delete table S1 in the manuscript. The authors have included the same table as Table S1 in the supplementary data file.
3) Please replace "the the combination" with "the combination" on Ln # 30 of page # 1.
4) Please replace "develops" with "develop" on Ln# 44 of page # 1.
5) Please rewrite the sentence "plant-derived phytochemicals......with very low systemic toxicity" on Ln # 47 of page # 2.
6) Please rewrite the sentence "These high concentrations are.....respectively" on Ln # 55 of page # 2.
7) Please replace "effect of" with "effect on" on Ln # 63 of page # 2.
8) Please correct "fewer side effects omparing to" to "fewer side effects compared to" on Ln # 98 of page # 3.
9) Could authors please follow consistent pattern to denote the combination treatment either as L30I40 or LUT+I3C.
10) Please replace "Tumor size was continue measured" with "Tumor size was continuously measured" on Ln # 383 of page # 11
11) Please proofread the manuscript for syntax errors.

Reviewer 2 Report

The authors described that combination therapy of luteolin (LUT) and indole-3-carbinol (I3C) could inhibit ERα-positive breast cancer cell. This study is interesting; however, there are some issues to resolve.

  1. In vitro study, 30 μM of LUT and 40 μM of I3C were used; however, these concentration was higher than the plasma concentration of these chemicals. Why did the authors use these concentration?

  1. The authors concluded that the combination of LUT and L3C could inhibit the cell proliferation only in ERα-positive breast cancer cell. However, only one cell line (MCF-7) was used as ER-positive breast cancer cell. Cell viability in vitro and tumor growth inhibition in vivo should be confirmed using other ERα-positive breast cancer cell line.

  1. Twelve phytochemicals were used in this study, and LUT and I3C was the best combination for ER-positive breast cancer cells. The authors should show the data of other combinations. I recommend the authors to add a supplemental table of combination index (CI) for all combinations.

  1. In the table S1, unit (μM?) is required.

  1. What did SERD (P6L13) and CUR (P7L18) stand for?

Reviewer 3 Report

The present manuscript by Wang and colleagus is based on previously published data that luteolin (LUT) and indole-3-carbinol (I3C) in combination has synergistic effects on ERalpha expression (Wang TT et al, J Nutr Biochem, 2006) and thereby may have anti-proliferative effects on ERa-positive breast cancer cells. Although combination of treatments that alone fail to reach efficacy is of clinical relevance, this manuscript fails to address several important issues that would aid in the interpretation of the data. In its present form the manuscript does not provide significant new knowledge that can be put in relation to what is already known in the field. In addition, numerous grammatical errors and typos throughout the manuscript makes it hard for the reader to assess the conclusion of the observations. In particular, the abstract should be totally rewritten.

Major issues:

  1. Proposing new combination of treatments at very high concentrations is controversial due to the high risk of off-target effects. The authors propose that the L30I40 combination (at high µM concentrations) decreases MCF7 cell proliferation (using an ATP-based assay) and increases apoptosis, and that this is ERa-dependent. Since many cell types, other than breast cells, express ERa, the authors should include such other cell types in their key experiments. It can be that all ERa+ cells undergo apoptosis rendering the L30I40 treatment clinically irrelevant. Similarly, the mice should be better analyzed for such effects. Did the mouse body weight get reduced? Were other ERa+ tissues (eg brain or reproductive tract) affected, etc?

  1. Lut at 30 µM inhibited growth rate in the presence of as low as 0.01nM E2 (Fig 3A). However, at the same time, the authors claim that LUT at 30µM does interfere with ERa activity (Fig 3C). This is contradictory and should be explained.

  1. Since the authors suggest that L30I40 has a strong interaction with the estrogen signaling, they should also investigate this interaction in their in vivo model by ovariectomizing the mice (reflecting the postmenopausal situation).

  1. What about interaction with AhR signaling? The authors discuss this, but there is no clear conclusion. Experiments that incorporate AhR signaling in L30I40 treatments should be included.

Round 2

Reviewer 1 Report

The objective of this study is to screen different phytochemicals to identify a combination which could exert synergistic inhibitory effect on breast cancer cells and xenografts. The authors have reported that combination of LUT+I3C have exerted synergistic inhibitory effect on ER+ breast cancer cell line MCF-7 cells and xenografts but not on TNBC cell lines. It is interesting data and the authors have addressed several concerns but the manuscript could not be considered for publication for following reasons.

Major concerns:

1) The take home message from the authors findings is that combination of LUT+I3C have exerted synergistic inhibitory effect on ER+ breast cancer cell line MCF-7 cells and xenografts. It is very interesting conclusion however the authors findings were based on only one ER+ breast cancer cell line (MCF-7) and is a major concern. The authors responded that they intend to repeat this experiment with inclusion of other ER+ cell lines at later time however it is very important to study whether the findings are specific to MCF-7 cell line alone or other ER+ positive breast cancer cells. It would give more credibility to authors findings if authors could include data from more than one ER+ breast cell line.

2) The data from various experiments in the manuscript has shown I3C alone did not have any effect on cell proliferation, cell cycle dynamics, cell viability, apoptosis, E2 stimulated cell growth in MCF-7 cells however the combination treatment of LUT+I3C has dramatic effect on aforementioned dynamics on MCF-7 cells. Could authors please include data on possible mechanism of action on how LUT+I3C combination has dramatic effects compared to single agent treatment.

3) The authors have proposed that combination of L30I40 inhibits breast cancer cell proliferation by inducing G1 cell cycle arrest and induces apoptosis. However L30I40 combination has arrested cells at G1 phase significantly at 24 only but not at 48h. Could authors please comment on possible reasons for lack of effect at 48h in cells treated with L30I40 compared to other Tx groups.

Reviewer 2 Report

The authors replied to my comments. I can agree to the authors’ responses.

Reviewer 3 Report

The authors have not adequately addressed the raised issues in the previous review. Mainly:

The authors did not include other ERa+ cells in their experiments as requested. The EAhy926 cell line is included in figure panels 1F and 1G, but what about effects on apoptosis, CDKs, CCD1, etc, in these cells? This is of importance for significance of the study. Also, what about the effect of the combination on other ER+ cells. The authors have added that they want to do such experiments in the future. However, it is already warranted in the present study. Such experiments must be included as a minimum - at least to compare with other ER+ breast cancer cells.

The effect on bodyweight of the mice is of relevance to the interpretation of the data and should be included as a supplementary figure. I cannot find any new information on lines 184-186 as the authors claim.

It is a clear limitation that the authors have not analyzed any other ER+ tissues in the mice. What was the reason for this, since they test a drug combination that apparently severely interferes with estrogen signaling?

Since E2 must be supplemented to the MCF7 xenograft model it raises questions to the relevance of using this model here. The authors must discuss drawbacks/limitations with the MCF7 xenograft model for their study.

Similarly to above, is the present study only relevant for women of fertile age? The authors must discuss this. What about the post-menopausal situation?

The question still remains regarding figure 3A & 3C. The authors write that Lut alone only works in the absence of E2 (Fig 3C). However in Fig 3A Lut inhibited E2-stimulated growth. This is contradictory and provides doubt to the soundness of the experiments. Also, statistics are missing in Fig 3A, B.

The manuscript has numerous grammatical errors throughout. An English native speaking person should go through and correct these errors. In addition, the abstract is still poorly written due to grammatical errors, which makes it hard to understand the purpose and outcome. It should be rewritten.